



# Tempo-spatial variation of the late Mesozoic volcanism in Southeast
China testing the western Paleo-Pacific Plate subduction models

**Xianghui Li[1, 2] [*]   Yongxiang Li[1]   Jingyu Wang[1]   Chaokai Zhang[1]   Yin Wang[3]   Ling Liu[3]**
[1]*State Key Laboratory for Mineral Deposits Research, School of Earth Sciences and Engineering,*
*Nanjing University, Nanjing 210023 China. Email: leeschhui@126.com*
[2]*Institute of Sedimentary Geology, Chengdu University of Technology, Chengdu 610059, China*
[3]*East China Mineral Exploration and Development Bureau, Nanjing 210007, China*





**ABSTRACT**
The westward subduction of Paleo-Pacific plate (PPP) played a governing role in tectonic evolution
of East Asia. Although various PPP subduction models have been proposed, the subduction age and
dynamical process of the PPP remain controversial. In this study, we investigate the geochronology
of extrusive rocks and tempo-spatial variations of the late Mesozoic volcanism in Southeast China.
We reported zircon U-Pb ages of new 48 extrusive rock samples in the Shi-Hang tectonic zone.
Together with the published data, ages of ~300 rock samples from ~40 lithostratigraphic units were
compiled, potentially documenting a relatively complete history and spatial distribution of the late
Mesozoic volcanism in Southeast China. The results show that the extrusive rocks spanned ~95 Myr
(177-82 Ma), but dominantly ~70 Myr (160-90 Ma), with two main age populations of 145-125 Ma
and 105-95 Ma. We propose that these ages represent the intervals of the Yanshanian volcanism in
Southeast China and the western subduction of the PPP, within which two intensive volcanic
eruptional pulses happened. Spatially, the age geographic pattern of extrusive rocks is both the oldest
and youngest age clusters occurring in the CZ and the younger intensive group in the SHTB,
indicating that the late Mesozoic volcanism migrated northwestly from the coast to the inland prior
to ~145 Ma and subsequently retreated southeastly back to the coast. This migration pattern is
interpreted to result from a northwestward subduction followed by a southeastward rollback or
retreat of the PPP.
**Keywords:** geochronology; tempo-spatial variation; volcanism; late Mesozoic; Southeast China;
Paleo-Pacific Plate





## 1. Introduction

It is generally believed that an Andean-type active continental margin had been developed

during the late Mesozoic in eastern Eurasia along which the Paleo-Pacific plate (PPP) subducted

beneath the East Asia (e.g., Taylor and Hayes, 1983; Faure and Natal'in, 1992; Charvet et al., 1994;

Zhou and Li, 2000; Chen et al., 2005; Liu et al., 2017; Li et al., 2019a). The subduction has exerted

profound impacts in Southeast (SE) China (e.g., Taylor and Hayes, 1983; Zhou and Li, 2000; Li CL

et al., 2014; Li JH et al., 2014; Jiang et al., 2015; Liu et al., 2016; ) and many other parts of East Asia

(e.g., Stepashko, 2006; Wu et al., 2007; Choi and Lee, 2011; Zhang et al., 2011; Sun et al., 2013;


2015; Dong et al., 2016; Liu et al., 2017), as indicated by the pervasive crustal deformation


associated with the Yanshanian orogeny (e.g., Lapierre et al., 1997; Li, 2000; Zhou and Li, 2000)


and the widespread magmatism (e.g., Zhou et al., 2006; Sun et al., 2007).


While the overall tectonic setting of the western Pacific in the late Mesozoic is generally


accepted, details such as the direction and angle of the PPP subduction remain controversial (e.g.,


Zhou and Li, 2000; Li and Li, 2007; Sun et al., 2007; Wang et al., 2011; Liu et al., 2012, 2014, 2016;


Zheng et al., 2017; Jia et al., 2018). Several tectonic models have been put forward to explain the


subduction process or geodynamics. (summary see Jiang et al., 2015; Li et al., 2018). Typical models


are: normal subduction (e.g., Lapierre et al. 1997), shallow subduction (e.g., Zhou and Li 2000; Jiang


et al. 2009), flat-slab subduction (Li and Li 2007), and subduction initiation in the Permian (e.g., Li


and Li 2007 Li et al., 2006; Knittel et al., 2010; Li et al., 2012a, 2012b), Middle Jurassic (e.g., Zhou


and Li 2000; Li et al., 2007; Jiang et al. 2009), and Early Cretaceous (e.g., Chen et al. 2008; Liu et al.,


2012, 2014). These models were postulated mostly based on the early sparse (bulk K-Ar, Ar-Ar,


Rb-Sr) dating data and / or from local and a limited number of samples in each individual article.


One way to test the relevant models is to investigate the spatial and temporal variations of the


widespread volcanism during the late Mesozoic in SE China. This effort is facilitated by the existing






abundant chronological, geochemical, and isotopic data of magmatic rocks from SE China. For the
volcanism responded to the PPP subduction, different time intervals and various episodes / cycles /
periods of volcanism have been proposed (e.g., Li et al., 1989; Guo et al., 2012; Li CL et al., 2014;
Liu et al., 2012, 2014, 2016; Jiang et al., 2015; Ji et al., 2018; Zhang et al., 2018; Yang et al., 2018;
Zhang et al., 2019). However, these different views were generally based on separate and often
limited datasets that were commonly from only several to a dozen of samples from a local region
such as a mining field, or a province, or at most from a relatively wide area of two neighboring
provinces. It is essential to obtain spatially more comprehensive datasets from different parts of SE
China and also temporally more expanded datasets from sedimentary basin archives that can
document the relatively complete volcanic history to achieve a holistic understanding of the late
Mesozoic volcanism and geodynamics in SE China.

In this study, we investigate the geochronology of extrusive rocks in the middle and northern

Shi-Hang tectonic belt (SHTB. e.g., Gilder et al., 1996; Jiang et al., 2011; Yang et al., 2012). The
SHTB contains thick sedimentary strata, which are interbedded with extrusive rocks, and thus has the
advantage of providing a more complete stratigraphic archive that preserves more complete and
recognizable volcanic events. We also compile the published zircon U-Pb isotope geochronological
data of extrusive rocks from the entire SE China. Obviously, ages of the extrusive rocks can
constrain the geochronology of the initiation, evolution, and termination of the late Mesozoic
volcanism in SE China, i.e. can also help date and better understand the slab subduction between the
Asian continent and the PPP in East Asia (e.g., Gilder et al., 1991, 1996). Specifically, we analyze
the temporal evolution and the geographical distribution of the late Mesozoic volcanism, whereby
the dynamics and process of the PPP subduction can be examined.
**2. Geological setting**

The South China Block comprises the Yangtze Block and Cathaysia Block. The Yangtze Block

has an Archean to Proterozoic basement, whereas the Cathaysia Block has a Proterozoic basement.

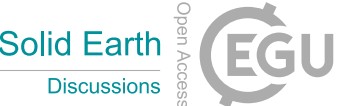

Yangtze and Cathaysia blocks amalgamated during the early Neoproterozoic Orogeny (e.g., Zhao
and Cawood, 1999; Wang et al., 2006; Zheng et al., 2007; Li et al., 2009), forming the Jiangnan
orogen. A cover sequence of marine strata from the late Neoproterozoic to the Paleozoic was
accumulated on the united South China Block that subsequently underwent the Caledonian orogeny
(or the Guangxi movement) in the early Paleozoic (e.g., Guo et al., 1989; Qiu et al., 2000; Charvet et
al., 2010) and the Indosinian orogeny in the early Mesozoic (e.g., Carter et al., 2001; Lepvrier et al.,

2004).

The major Jiangshan-Shaoxing suture zone separating the Yangtze and Cathaysia blocks (e.g.,
Jiang et al., 2011; Yang et al., 2012) had been reactivated during the Indosinian and Yanshanian
movements. During the late Mesozoic Yanshanian, the Andean-type convergent margin was
developed along the SE China following the subduction of the PPP. A series of NE-striking back-arc
basins associated with widespread and large-scale magmatism were produced (e.g., Zhou and Li,
2000; Li and Li, 2007; Liu et al., 2014, 2016; Xie et al., 2017; Yang et al., 2017). Since the
deposition in these basins was concomitant with volcanism, it is fairly common that the sedimentary
successions are interbedded with volcanic rocks. On the basis of the abundance of volcanic rocks in
the strata, these basins can be grouped into three types (Fig. 1): volcanic (-dominated),
volcanic-sedimentary, and sedimentary (e.g., Chen et al., 2005; Shu et al., 2009). These three types
of basins are roughly separated by two NE-striking fault zones: the Jiangshan-Shaoxing fault zone
and the Zhenghe-Dapu fault zone (Fig. 1). The volcanic basins occur SE to the Zhenghe-Dapu fault
zone and were formed on the magmatic arc along the coastline, i.e., the Coastal zone (CZ). The
volcanic-sedimentary basins occur in the SHTB confined between the two fault zones, and volcanic
rocks are typically interbedded and / or intercalated with sedimentary strata. Nevertheless, the late
Mesozoic volcanic rocks are almost absent east to the Yujiang-Yudu fault zone in sedimentary basins
and western SHTB basins (Fig. 1).
The large-scale magmatism is evidenced by the occurrence of granitic plutons in both the SHTB





and the CZ stretching over 1000 km along the coastal SE China. These granitic plutons intruded into
the Precambrian basement and the overlying Paleozoic strata during the Middle Jurassic - Early
Cretaceous (e.g., Jiang et al., 2011; Yang et al., 2012). The intrusions mainly occur as A-type and /
or I-type granitic rocks, and together with huge volcanic rocks, strongly support the model of the
western subduction of the PPP (e.g., Zhao et al., 2016; Jiang et al., 2011; Yang et al., 2012; Xie et al.,
2017; Yang et al., 2017).

### 3. Material and methods

A total of 48 extrusive rock samples were collected from about 20 lithostratigraphic formations
(supplementary data Table RD1) in 11 basins / regions within the main SHTB to obtain new zircon
U-Pb isotope ages (L1-L10 in Fig. 1; supplementary data Figs. RD1-RD3 and Table RD1). The
extrusive rock specimens are volcanic and pyroclastic rocks that are interbedded and intercalated
with the sedimentary strata, in which sampling horizons and associated lithologies are marked in the
supplementary data figures RD4-RD12. These samples were collected from volcanic layers in the
main type sections of typical basins in SE China (supplementary data Fig. RD4-RD12). In general,
3-4 rock samples were taken at lower/base, middle and upper/top part when a lithostratigraphic unit
has multiple volcanic horizons or a volcanic layer is over 100-200 m thick (see supplementary data
Table RD1). The locations of these samples were determined with a GPS device and are marked on
the geological maps (supplementary data Figs. RD1-RD3 and Table RD1).
Zircon grains were separated using the conventional heavy liquid and magnetic techniques.
Single zircon grains were handpicked and mounted on adhesive tapes, embedded in epoxy resin, and
then polished to about half to one-third of their thickness and photographed in both reflected and
transmitted light. Cathodoluminescence (CL) images were taken at the State Key Laboratory for
Mineral Deposits Research, School of Earth Sciences and Engineering, Nanjing University, to
examine the internal structures of single zircon grains before U-Pb isotope analysis.





LA-ICP-MS, U-Th-Pb analyses of single zircon grains were performed on a Nd of YAG 213
laser ablation system (Agilent 7500a, New Wave Research, U.S.A.) coupled with VG PQ Excell
ICP-MS, which is housed in the State Key Laboratory for Mineral Deposits Research, Nanjing
University. General ablation time is ca. 60 s and the ablation pit diameter is at 25-35 μm. The
ablation repetition rate is 5 Hz with the incident pulse energy of about 10~20 J/cm$^2$. Calibrations of
mass fractionation were made using the index sample GEMOC/GJ (608 Ma). In each experiment, a
total of 11 to 21 zircon grains were measured, among which 8 to 18 grains yield concordant age data.
Prior to each experiment, the standard GJ-1 and Mud Tank samples were measured. Other
measurements follow the methods described by Jackson et al. (2004). Analyses of Mud Tank sample
yielded a weighted $^{206}$Pb/$^{238}$U age of 726±10 Ma~737±5 Ma (2σ), which is in good agreement with
the recommended value (TIMS age =732±5 Ma, Black and Gulson, 1978).
Data reduction, isotope ratio, age calculation, and Pb correction were conducted with the
GLITTER software using Zircon 91500 as an external standard. Data processing and plotting were
executed with the Isoplot 3.23 programs (Ludwig, 2001). The uncertainties of age results are quoted
at 1σ confidence level, whereas errors for weighted mean ages are quoted at 2σ.
It is worth noting that those aged samples of mafic dykes, basalts and gabbros were not herein
compiled for the analysis of volcanic temporal-spatial variation in SE China. This is because: 1)
among the magmatic rocks, gabbros and basalts are rare, and diorites and andesites are even less
common in South China (Zhou and Li, 2000), leading to a weak significance in statistic of the
volcanic samples; 2) those published ages of the dykes, basalts and gabbros were mainly measured
using different (Ar-Ar, K-Ar, Rb-Sr) isotopic methods (e.g., Li, et al., 1989; Chen et al., 2008b;
Wang et al., 2008; Meng et al., 2012), likely causing chaos of real ages; 3) it is difficult to obtain a
good isotopic age for mafic rocks, and particularly, the bulk (basalt) samples ages by K-Ar, Ar-Ar,
and Rb-Sr are ~ 10-20 Ma younger than those by zircon U-Pb isotopes (Li et al., 2019b); and 4)
some basalts are of the Indosinian orogeny age, instead of the Yanshanian orogeny.



## 4. Results

### 4.1 Uncertainty of zircon U-Pb ages

It is necessary to first evaluate the uncertainty of the new age results and other cited age data. The uncertainty depends on three aspects, i.e. origin of zircon, precision, and accuracy (Schoene et al., 2013).

For the origin, all zircons used in this work were microscopically evaluated with CL to ensure that laser ablation positions of zircons are away from the nucleus, cracks, and inclusions. CL images manifest the growth rings. In the concordant 636 zircons of this work, 20 grains (3.1%) are 0.1-1.0 in Th/U ratio, 615 (96.7%) are 1.0-10.0 (Table 1). Th/U ratios of 3539 zircons can be available in the age data from published references. Together with published data and this work, 1766 zircon grains (42.3%) are 0.1-1.0 in Th/U ratio, 2394 grains (57.3%) are 1.0-10.0, 14 grains (0.3%) are > 10.0, and only one is less than 0.1 (Table 1). CL images and Th/U ratios of this work combined the collected data demonstrate that predominant (>99.9%), if not all, zircons are magmatic origin.

Precision and accuracy uncertainties produced during LA-ICP-MS zircon U-Pb dating have been more and more concerned (e.g., Klötzli et al., 2009; Solari et al., 2010; Li et al., 2015) and come from multiple sources, including the isotopic ratio measurements, the fractionation factor calculation using an external standard, the common lead correction, the external standards, and the data reduction (Li et al., 2015). According to the suggested ~4% (2σ) of precision and accuracy (Li et al., 2015), we used the ~2% and ~2-4% (1σ) to evaluate uncertainties of extrusive rock ages.

A total of 48 rock samples were respectively weighed in mean from 636 concordant zircon U-Pb ages in this work (supplementary data Table RD1 and RD2). In the samples, 46 (95.8%) have a <3 million years (Myr) error in 1σ, in which 36 (75%) samples have <2 Myr error in 1σ; 41 samples (85.4%) have <2.0% (error / age) deviation, and 7 (14.6%) have 2-4% deviation (Table 1). Similar percentages of sample error and age deviation are comparative with those single zircons analyzed in this work (Table 1).

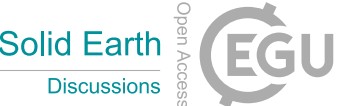

For zircons from the published data, the literature often provides CL images of zircons showing
quite similar nature in source and error. For the zircon U-Pb ages from the previous studies, we
carefully examine the experiments described in the literature, re-analyze the concordant ages, and
eliminate those that are not concordant and / or greater than ~5% in age deviation (error / age) as
well as ages with distinct inheritance, which were not discarded by the original authors. This
scrutinizing procedure allows us to identify reliable U-Pb age data from 188 volcanic rock samples
from the SHTB and from 103 volcanic rock samples from the CZ (supplementary data Table RD2
and RD3). Then, results show that in the combined 291 samples, 246 samples (85.5% = 246/291) are
<2 Myr in 1σ error of age and 39 (13.4 %) are 2-4 Myr; and 264 samples (90.7%) are <2.0% age
deviation and 25 (8.6%) are 2-4 Myr deviation in age (Table 1). Closely, total concordant single
zircons 4639 are similar in percentages of 1σ error and age deviation with the weighed-mean age
samples (Table 1).
The above relatively low errors in 1σ and deviation of age indicate that samples of both this and
previous work have highly proportional age results (> ~95%) with fine precision.
Systematic biases often dominate uncertainty in comparisons between dating methods and
between laboratories (Schoene et al., 2013). For measurements of our zircon samples, the internal
systematic 2σ error is less than 3%, which has been verified by reproductive measurements of Mud
Tank sample (see Section 3). These systematic biases were mostly met for those zircons from the
references. Therefore, small internal systematic 2σ errors allow our zircon date results to be a
moderate accuracy in geochronological application.
The internal systematic conditions are same for weighted mean dates of individual samples from
both this and previous work (ref. and comp. to supplementary data Table RD1-RD3). Compiled
zircons are predominantly single dates generally within less than 2 Myr in 1σ errors (<3% biases) for
the Late Jurassic – Early Cretaceous volcanic rocks. The dates are to great degree consistent with the
biostratigraphy of pollens-spores, plants, ostracods, and conchostracans in the volcanic-sedimentary





basins, SHTB (e.g., Chen and Shen, 1982; Sha, 1990; Jiang et al., 1993; Chen, 2000).
In summary, the zircon origin and the age precision and accuracy indicate the sample
weighed-mean ages have relatively low uncertainty and they are eligible for investigating the
eruption geochronology of extrusive rocks in SE China.
**4.2 U-Pb age spectra of extrusive rocks**
Spot analyzing results of this work show that 48 samples have a wide range of (concordant
$^{206}Pb/^{238}U$) weighed-mean ages from 162 Ma to 92 Ma (green histogram, Fig. 2a), in which four age
populations, ~162-150 Ma, ~144-112 Ma, ~112-102 Ma, and ~102-92 Ma, can be observed. In these
populations, two peaks of weighted mean ages are regressed as 133.3 ± 1.5 Ma and 97.2 ± 1.1 Ma,
respectively (Fig. 2a). In addition, 636 concordant single zircons from the samples show similar wide
age range (166 to 92 Ma) with four age populations and two age peaks (Fig. 2a).
Combining our new results with the published age data from the main SHTB (e.g., Wu et al.,
2011a, b; Wu and Wu, 2013; Liu et al., 2012, 2014, 2016; Li CL et al., 2014; Li JH et al., 2014; Ma
et al., 2016; Wang et al., 2016; Shu et al., 2017. Locations M1-M22, Table RD1 and Fig. 1) yields a
similar age pattern (Fig. 2b). A total of 188 rock samples show that the weighed-mean age range
from 177 Ma to 92 Ma with four main populations ~162-144 Ma, ~144-128 Ma, ~128-104 Ma, and
104-92 Ma and two age peaks 136.11 ± 0.38 Ma and 100.0 ± 1.0 Ma (Fig. 2b). Also, a total of 2593
single zircons from the SHTB show the concordant $^{206}Pb/^{238}U$ age ranging from 180 Ma to 92 Ma
with four age populations and two age peaks 132.07 ± 0.17 Ma and 101.26 ± 0.23 Ma (Fig. 2b).
The published data of 103 rock samples from the CZ (for Locations N1-N21, see Fig. 1 and
supplementary data Table RD1 and RD3. Chen et al., 2008; Li et al., 2009; Guo et al., 2012; Li CL et
al., 2014; Liu et al., 2012, 2016; Zhang et al., 2018) show a wide weighed-mean ages ranging from
174 Ma to 82 Ma, five main age populations of ~174-150 Ma, ~150-126 Ma, ~126-102 Ma, ~102-92
Ma, and ~92-82 Ma, and two age peaks of 130.96 ± 0.87 Ma and 98.13 ± 0.55 Ma (Fig. 2c), similar



to those from the SHTB (comp. Fig. 2b and 2c). The 1942 single zircons from the 103 samples also
display the same range of concordant $^{206}Pb/^{238}U$ ages (Fig. 2c; supplementary data Table RD3) with
similar five main age populations (~180-146 Ma, ~146-126 Ma, ~126-102 Ma, ~102-94 Ma, and
~94-76 Ma) and two age peaks (131.04 ± 0.32 Ma and 99.08 ± 0.32 Ma. Fig. 2c).
Further combined and optimized age data of 291 extrusive rock samples of over 40
lithostratigraphic units in both SHTB and CZ illustrate that sample weighed-mean ages mainly vary
between 177 Ma and 82 Ma, which can be classified as five populations: ~178-145 Ma, ~145-125
Ma, ~125-105 Ma, ~105-95 Ma, and ~95-82 Ma (Fig. 3). Of the populations, two peaks are at 133.87
± 0.5 Ma (93 samples, 138-130 Ma, MSWD = 3.7) and 98.19 ± 0.47 Ma (25 samples, 100-96 Ma,
MSWD = 1.14), respectively. The compilation of age data from all the 4639 concordant single
zircons shows that the $^{206}Pb/^{238}U$ ages range between ~180 Ma and ~76 Ma with five populations of
~180-145 Ma, ~145-125 Ma, ~125-105 Ma, ~105-95 Ma, and ~95-76 Ma and two age peaks at
132.90 ± 0.14 Ma and 99.86 ± 0.19 Ma (Fig. 3).
**5. Discussion**
**5.1 Temporal evolution of volcanism**
The late Mesozoic extrusive rocks are widespread in SE China and their dating has been
conducted extensively. In early times, they have been roughly dated as the (Late) Jurassic and (to the
Late) Cretaceous by the confinement of interbedded / intercalated terrestrial fossil-bearing
sedimentary strata, and the ages are quite crude. Later on, Rb-Sr, K-Ar, and Ar-Ar dating of
bulk-dominated samples yielded ages of ~150-65 Ma with large age uncertainties in the 1980s-1990s
(e.g., Hu et al., 1982; Li et al., 1989; Feng et a., 1993; Zhang, 1997), much younger than the earlier
rough estimates, and ~10-20 Myr younger than the zircon U-Pb isotope ages on average (Li et al.,
2019b).
In the recent decade, though zircon U-Pb age data of the igneous rocks have been reported, rock
samples in individual references were taken from separate locations resulting in different age



interpretations of volcanic eruption in SE China, and a relative concurrent viewpoint has not been
reached. Multiple volcanic age durations are available at different locations or regions, such as
145-129 Ma, 143-98 Ma, and 140-118 Ma in eastern and northwestern Zhejiang (Liu et al., 2014),
140-88 Ma and 136-129 Ma in southeastern (Liu et al., 2012) and central Zhejiang (Li JH et al.,
2014), 168-95 Ma in northeastern Guangdong and southeastern Fujian (Guo et al., 2012), 162-130
Ma from two locations in Fujian (Li et al., 2009), 160-99 Ma from northern Fujian (Liu et al., 2016),
and 112-99 Ma from Zijingshan Mineral Field of Fujian (Jiang et al., 2013, 2015). Obviously, these
ages are incomplete and intermittent, and cannot individually reveal the age of volcanism in the
entire SE China.

To investigate the geochronology of extrusive rocks, we conducted zircon U-Pb age analysis in

the SHTB and combined the published data from both SHTB and CZ. Then relatively high precise
and representative dating results are obtained in entire SE China: the combined and optimized ages
from 291 rock samples (4639 concordant zircons) range from ~177 Ma to ~82 Ma (mainly 160-90
Ma). As we know, the U-Pb isotope ages of zircons represent the cease time of the crystalline zircon
formation when volcanic eruption, therefore, we propose that the age range above is an eligible
representation for the duration of volcanism in SE China. That means, the volcanism could have
initiated at the late Toarcian (~177 Ma) of the late Early Jurassic and terminated at the early
Campanian (~82 Ma) of the Late Cretaceous, and it has a ~95 Myr duration, which shows little
discrepancy with those of the single zircon ages (Fig. 3). On the other hand, the volcanism occurred
chiefly during the interval of the Late Jurassic-Early Cretaceous (160-90 Ma = 70 Myr) when only
several samples with ages of pre-160 Ma and post-90 Ma are regarded (e.g., Chen et al., 2007; Guo
et al., 2012; Liu et al., 2012). When one considers the relationship of the magmatism originated by
the PPP subduction (details see section 5.3), the above age range and duration are also suggested to
represent the westward subduction time of the PPP during the Yanshanian orogeny in East and SE
Asia.


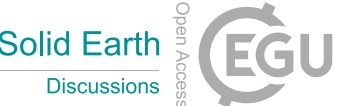

It is noted that among the compiled single zircon U-Pb ages of extrusive rocks, the oldest one is
from the Maonong Formation (Fm) in Songyang Basin in Zhejiang. The weighted mean age is 177.4
± 1.0 Ma for the sample MN01 (location M14. Liu et al., 2012). In addition, a weighted mean age of
180 ± 4 Ma from the same horizon (Chen et al., 2007) has also been reported despite that the error is
relatively large, up to 6-8 Myr.
Similarly, variable youngest ages of volcanic rocks are reported. The weighted mean age of 82.5
± 1.0 Ma of the sample ZJ23 (location N2. Chen et al., 2008) from the Taozu section of eastern
Zhejiang could be the youngest age. One zircon grain from the section is dated at 74 ± 0.6 Ma and
five zircon grains yield concordant ages of 76 ± 0.6 Ma from the same sample (Table RD3. Chen et
al., 2008), suggesting that it is possible the termination of volcanism was ~5 Myr younger than 82.5
Ma.
As we can see from section 4.2, five age populations of extrusive rocks were recognized. We
herein suggest age populations as corresponding five stages / pulses of volcanism in SE China, and
the two populations ~145-125 Ma and ~105-95 Ma could be the intervals of intensive volcanism in
SE China, respectively. The most extensively volcanic eruption episode (145-25 Ma) seems to
correspond to the period of rapid increase in the magmatic flux of both the Mid-ocean ridge and
Large Igneous Provinces (Coffin & Eldholm, 1994) during the late Late Jurassic - early Early
Cretaceous (Fig. 4) although the relationship between them remains unclear.
**5.2 Spatial pattern of volcanism**
Though it is well-known that the late Mesozoic magmatic rocks are widespread in SE China, the
previous volcanic distributions are to some degree out of date as those ages contain large errors with
low preciseness and accuracy by bulk isotope dating (e.g., Li et al., 1989; Wang et al., 2000; Zhou
and Li, 2000; Chen et al., 2008b) and detailed age distribution patterns by precise age constraints
have not been outlined yet. To delineate the spatial variation and migration process of the late



Mesozoic volcanism in SE China, we sketched two age distribution maps of extrusive rocks showing
the initial and terminal ages of volcanism (Figs. 5 and 6).
Firstly, we identified the initial ages of extrusive rocks. The initial age is defined as the earliest
age of volcanic eruption in a location, a basin, and / or a region marked as capital letters L, M and N
with numbers in figure 1. Three age boundaries ~163 Ma, ~145 Ma, and ~125 Ma are chosen to
divide the initial ages into four intervals: 180-163 Ma, 163-145 Ma, 145-125 Ma, and 125-76 Ma,
which are somewhat different from the classification of volcanic pulses in section 5.1. Actually, the
three age boundaries ordinarily and clearly correspond to those of the Middle and Late Jurassic, the
Late Jurassic and Early Cretaceous, and the early and mid-Cretaceous, respectively. We used the
boundary age 163 Ma as a separate boundary within the first period of the volcanism because it
represents the initiation time of the first Yanshanian orogenic episode in East and SE Asia and the
corresponding stratal boundary is marked by an unconformity (e.g., Yu et al., 2003; Shu et al., 2009).
The boundary between the Upper Jurassic and the Lower Cretaceous is also represented by a widely
observed unconformity (e.g., Yu et al., 2003; Shu et al., 2009) and the intensification of volcanism in
SE China (Fig. 3). As there are fewer samples with ages of < 125 Ma and the age boundary at ~125
Ma marks the rapid waning of volcanism (Fig. 3), we combined the three periods 125-105 Ma,
105-95 Ma, and <95 Ma of volcanism as one initial recognition.
Then, isolines ages are drown by the boundary age 163 Ma, 145 Ma, and 125 Ma, separately.
Interpolation ages are used to confine the zones when there are no exact ages same as the boundary
age occur in the map. Plotting the initial ages in the geographical map shows four zones of volcanism
in SE China (Fig. 5). Zone 1 (177-163 Ma) marks areas where initial volcanic eruption locally occurs
in the northernmost Guangdong and neighboring southern Fujian and northeastern Fujian in the CZ
and at one location of southwestern Zhejiang (M14, Songyang, Liu et al., 2012) in SHTB. Zone 2
(163-145 Ma) delineates areas where initial volcanic eruption occurs around Zone 1 in southern and
northeastern Fujian in the CZ, and half extends into the SHTB (Fig. 5). Zone 3 (145-125 Ma) defines



regions where initial volcanic eruption chiefly and largely extends in the SHTB and mostly bounded
in west of the volcanic area, extending along the eastern Jiangxi, northwestern Fujian, and western
Zhejiang (Fig. 5). Zone 4 (125-90 Ma) locally occupies eastern Zhejiang and limited southeastern
Fujian in the middle-eastern CZ (Fig. 5). Same zones can be also recognized in the map made from
the single zircon U-Pb ages (comp. the supplementary data Fig. RD13), supporting the zonations of
the sample weighed-mean ages.
Secondly, five populations of 145-135 Ma, 135-125 Ma, 125-115 Ma, 115-95 Ma, and <95 Ma
are designed with a 10 Myr interval for the terminal volcanism in SE China, which are slightly
different from the initial ages (comp. Figs. 5 and 6). The age interval scheme is helpful to distinguish
the terminal volcanic distribution. This is because the main population ages are totally much younger
than 145 Ma and few samples are younger than 95 Ma, for which the main population isolines are
more readily made.
Similarly, isolines ages are drown by the boundary age 135 Ma, 125 Ma, 115 Ma, and 95 Ma,
separately, and interpolation ages are used to confine the zones when there are no exact ages in the
map. The isolines in the geographical map also shows five age zones for the terminal volcanism in
SE China (Fig. 6). Zone 1 (145-135 Ma) sparely occurs in the southern and northeastern Fujian,
similar to those of the initial age distribution. Zone 2 (135-125 Ma) mainly occurs in eastern Jiangxi,
western SHTB while partly surrounds the Zone 1; Zone 3 (125-105 Ma) distributes in the boundary
region of eastern Jiangxi and western Fujian and in middle Zhejiang in the SHTB. Zone 4 (105-95
Ma) appears in regions of the southern Fujian, middle Zhejiang in the eastern SHTB and CZ. Zone 5
(<95 Ma) sporadically displays in the eastern Fujian, eastern Zhejiang, and northern Guangdong in
the CZ. Same zonations can be classified in the map sketched by the single zircon U-Pb ages
(supplementary data Fig. RD14), verifying the zones of the sample weighed-mean ages in SE China.
Zonations of both initial and terminal volcanism indicate a distinct pattern of volcanic extrusion
in SE China (Figs. 5 and 6): the oldest ages in the CZ, the younger intensive age clusters in the



SHTB, and the youngest ones in the CZ. Detailed distributional patterns can be observed: 1) the
earliest appearance and earliest disappearance of extrusive rocks dominantly occur in southern and
northeastern Fujian in the CZ; 2) the most widespread distribution of 145-125 Ma extrusive rocks are
in eastern Jiangxi, western Zhejiang, and western Fujian in the SHTB; 3) the latest appearance and
latest disappearance mainly occur in eastern Zhejiang and eastern Fujian in the CZ. In summary, the
first appearance (initial volcanism) area is the first disappearance (terminal volcanism) region.
It is surprising that the zone 1 and / or 2 of both initial and terminal volcanism look like
thermal-dome patterns (Fig. 5 and 6) by exhumation and exposure that may be related to the regional
magmatic intrusion, likely misleading the migration of volcanism. However, the distribution pattern
is not dome-controlled because: 1) The data are derived from extrusive rocks, instead of intrusive
rocks; 2) it is impossible that a crater is over 200-300 km wide in diameter; 3) lots of agglomerates
representing craters were observed in a variety of strata at locations / basins out of Zone 1. For
instance, these agglomerates are widespread in basins of western Zhejiang (L1~L4; M9~M14),
eastern Jiangxi (L5~L7; M16~M18b), and western Fujian (L8~L10, M19~M22).
**5.3 Implication for the PPP Subduction**
It is accepted that the late Mesozoic (Yanshanian) magmatism was caused by the subduction of
the western PPP even though the subduction geodynamics, direction, and angle remain controversial
(e.g., Li and Li, 2007; Sun et al., 2007; Liu et al., 2012, 2014, 2016; Duan et al., 2017; Jia et al., 2018)
since the early propositions (e.g., Jahn, 1974; Lapierre et al., 1997; Zhou and Li, 2000). In the
subduction model, the magmatism was often attributed to the mantle-crust interaction, that is, the
geodynamic environment has been commonly regarded as an active continental margin related to the
subduction of the PPP under Eurasia (e.g., Engebretson et al., 1985; Maruyama and Seno, 1986;
Faure and Natal'in, 1992; Zhou and Li, 2000; Honza and Fujioka, 2004) and / or Northeast Asia (e.g.,
Stepashko, 2006; Wu et al., 2007; Choi and Lee, 2011; Zhang et al., 2011; Sun et al., 2013; 2015;
Dong et al., 2016; Liu et a., 2017) as well as SE China (e.g., Faure et al., 1996; Chen and Jahn, 1998;



Zhou and Li, 2000; Chen et al., 2005; Li et al., 2009; Liu et al., 2012, 2014, 2016; Jiang et al., 2013,
2015; Li CL et al., 2014; Li JH et al., 2014; Duan et al., 2017; Hong et al., 2018; Jia et al., 2018;
Zhang et al., 2018). Accordingly, the subduction angle (rollback hypothesis) and / or polarity change
are the crucial reference to geodynamics.

There are at least six main models put forward to explain the subduction direction and angles. 1)

In an early model, a so-called normal subduction of the PPP happened in the late Mesozoic by felsic
arc magmatism and continental olivine tholeiites (Lapierre et al., 1997). 2) PPP westward subducted
under the Andesite-type active margin in SE China since the Permian (e.g., Li et al., 2006; Knittel et
al., 2010; Li et al., 2012; Li et al., 2012). 3) The dip angle of the PPP subduction slab increased (low
to median angle) since the beginning of the Early Cretaceous, resulting in oceanward migration of
the magmatic zone to the coastal area (Zhou and Li, 2000). 4) A long-lasting, persistent
northwestward subduction between ~250 Ma and ~190 Ma with a subsequent retreat between ~180
Ma and ~155 Ma was proposed to explain the development of a broad (~1300-km-wide)
intracontinental orogen in South China (Li and Li, 2007). 5) The southwestward then northwestward
subducted in the late Mesozoic (180–125 Ma) (e.g., Sun et al., 2007); 6) The shallow subduction and
slab rollback took place during the Middle-Late Jurassic and late Early Cretaceous (e.g., Jiang et al.,
2009, 2015; He and Xu, 2012; Liu et al., 2014, 2016; Yang et al., 2018; Zhang et al., 2019).

However, these models were mostly based on two situations. One is that the authors mostly

employed the dating and geochemical data from unpublished and local reports in the 1980s-1990s,
which were mainly measured from (non-zircon) bulk samples using methods and techniques of
Ar-Ar, Rb-Sr, and Sm-Nd and others with less precision and accuracy. By the state of art at the time,
those data could have led to the misunderstanding of the model. Another situation is that well-dated
materials were mainly derived from a local mining field, a region, a province, or at most a boundary
area of two or three provinces. These two situations of imprecise ages and local material could have
resulted in incompleteness even mistaking on the PPP subduction process.



To examine the models of the PPP subduction directions and angles, we combined our new
zircon U-Pb dating works with lots of published ages and tried to analyze the tempo-spatial variation
of the late Mesozoic volcanism in SE China, which may shed new light on the PPP subduction. It is
worth noting that the association of the late Mesozoic volcanism in SE China with the western PPP
subduction has been demonstrated by numerous geochronological and geochemical studies of both
intrusive and extrusive rocks from variable locations (e.g., Yu et al., 2006; Jiang et al., 2011; Guo et
al., 2012; Yang et al., 2012; Liu et al., 2012, 2014, 2016; Jiang et al., 2015; Li WX et al., 2017; Shu
et al., 2017; Ji et al., 2018; Jia et al., 2018; Zhang et al., 2018). The age data of this study and those
compiled from previous work were derived from similar / same basins and / or locations (refer to Fig.
1 and supplementary data Table RD1), indicating the combined data have the same tectonic meaning.
That means the extrusive rock samples for age analysis used in this paper are eligible for the linkage
of the PPP subduction.
As shown in sections 5.1 and 5.2, zonations of both initial and terminal volcanism can be made
by age distribution of the late Mesozoic extrusive rocks, indicating a migration process of volcanic
extrusion in SE China. We proposed a volcanic process that took place in the following sequence
(Figs. 5, 6, and 7). Firstly, the volcanism occurred in northeastern Fujian and southern Fujian (Zone
1) of the CZ. It is noteworthy that a few zircon U-Pb isotope ages (interval ~195-180 Ma) of the
Early Jurassic from southernmost Jiangxi were recently published (e.g., Cen et al., 2016). These ages
belong to the late episode of the Indosinian orogeny. It likely indicates that Zone 1 can reach
southernmost Jiangxi if its relevance to the Yanshanian movement is verified. Secondly, the
magmatic extrusion happened in eastern Jiangxi and southwestern Fujian (Zone 2) in the main SHTB.
Then it appeared in northern Fujian and middle Zhejiang (Zone 3) in the SHTB. Finally, the volcanic
eruption had been transferred and emerged in eastern Zhejiang and at limited locations in
southeastern Fujian (Zone 4-5) in the CZ. The zonation and process of the volcanic extrusion suggest
that the volcanism first advanced northwestward and then retreated southeastward during the late

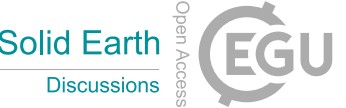

Mesozoic. The southeastward retreat of volcanism is also indicated by the change of the main
interval of intense volcanic extrusions, that is, the first interval (~145-125 Ma) of intensive
volcanism mainly occurred in eastern Jiangxi, western Fujian, and middle Zhejiang, indicating a
broad volcanism at the time; and the second one (105-95 Ma) mainly appeared in Tiantai area of
eastern Zhejiang and Fuqing-Dehua (southwest to Fuzhou) area of eastern Fujian (Figs. 5 and 6),
illustrating a constricted volcanism at last.

By the detailed characterization of the temporal and spatial variations of the late Mesozoic

volcanism from the much more comprehensive data of geochronology in SE China, we refined and
put forward a different single model of the subduction dynamics in western Pacific (Fig. 7). Based on
the migration pattern of volcanism (Fig. 5, 6, and 7), we propose that the PPP subducted
northwestward during the Middle-Late Jurassic (178-145 Ma. Fig. 7a) and the subduction slab then
rolled back or retreated southeastward during the main Early Cretaceous (145-95 Ma. Fig. 7b and 7c).
These would have led to the subsequent southeastward retreat of volcanism (Fig. 5) and to the
extension of back-arc by lithosphere foundering (Fig. 7b and 7c). The transfer in the migration
direction of volcanism from the northwestward to the southeastward may have occurred at ~145 Ma,
as evidenced by the great increase in the early Early Cretaceous age population (145-125 Ma) (Fig.
3), implying that the rollback of the PPP may have led to the Early Cretaceous lithospheric extension
(e.g., Li, 2000; Chen et al., 2008; Guo et al., 2012; Meng et al., 2012; Shu et al., 2017) and / or the
reactivation of the older NE-striking faults (e.g., Wang et al., 2013) in SE China (Fig. 7b and 7c).
Indeed, the rollback of the PPP has been proposed previously with the timing ranging from ~190 Ma
(e.g., Jiang YH et al., 2015; Cen et al., 2016) to ~90 Ma (e.g., Zhao et al., 2016). But the dominant
age interval for the initiation of the PPP rollback was ascribed to the 145-130 Ma (e.g., Li LM et al.,
2009; Yang et al., 2012; Li PJ et al., 2013; Li CL et al., 2014; Su et al., 2014; Li et al., 2017; Yang et
al., 2018). Combined with the widespread unconformity in SE China (e.g., Yu et al., 2003; Shu et al.,
2009), our results from the extrusive rocks indicate that ~145 Ma represents the initiation timing for





the rollback of the PPP subduction. Since the beginning of the Late Cretaceous (~ 95 Ma / 105 Ma),
the frontier of the PPP may be broken off and a new normal subduction was either re-established or
ceased (Fig. 7d). This alternation could have resulted in the fading of the magmatism and caused an
unconformity between the gravelly mollase Danxia Supergroup and the underlying Lishui
Supergroup in S China (Li et al., 2019b).
**6. Conclusions**

We analyzed weighed mean ages of 48 extrusive rock samples (total of 636 concordant single

zircons) from ~20 lithostratigraphic units at 11 localities in the SHTB. Published ages of 243 rock
samples (total of concordant 3662 zircons) from ~40 lithostratigraphic units in SE China are
compiled and re-examined. Based on a total of refined 291 sample ages (4639 concordant zircon
U-Pb ages) from this study and the published literatures, we propose that the late Mesozoic
volcanism in SE China initiated at ~177 Ma (late Toarcian of the late Early Jurassic) and terminated
at ~82 Ma (early Campanian of the Late Cretaceous), spanning an ~95 Myr interval (mainly ~70 Myr
= 160-90 Ma), during which two peak age populations at 145-125 Ma (the early Early Cretaceous)
and 105-95 Ma (the early Late Cretaceous) are interpreted to indicate the two pulses of intensive
volcanism. As the volcanism had been associated with the subduction in western Pacific, we suggest
that these age range and span represent the time of the Yanshanian subduction of the western PPP in
East and Southeast Asia.

The spatial change of the late Mesozoic volcanism is used to explore the linkage between the

volcanism and PPP subduction. A distinct pattern of volcanic extrusion ages in SE China is found:
both the oldest and youngest ages in the CZ and the intensive younger one in the SHTB. The
geographical distributions of the volcanic eruption ages reveal a migration process of magmatic
extrusion in SE China. The migration scenario of the volcanic extrusion can delineated as: the first
zone of volcanism occurred in northeastern Fujian and southern Fujian in the CZ, the second zone
moved northwestward to the eastern Jiangxi, western Fujian and western Zhejiang in the western



SHTB; then, the third zone retreated southeastward to the northwestern Fujian and the
middle-eastern Zhejiang in the SHTB; Finally, the last zone migrated to the eastern Zhejiang and the
middle-eastern Fujian in the CZ.
The tempo-spatial variations of the late Mesozoic extrusive migration indicate that the
volcanism first advanced northwestward and then retreated southeastward in SE China. This implies
that the PPP probably subducted northwestward during the Middle-Late Jurassic (177-145 Ma) and
the subduction slab then rolled back or retreated southeastward during the main Early Cretaceous
(145-95 Ma), leading to the subsequent southeastward retreat of volcanism. This change in the
migration direction of volcanism from the northwestward to the southeastward happened at ~145 Ma,
i.e., beginning of the Cretaceous, probably responsible for the Early Cretaceous lithospheric
extension behind the magmatic arc in South China.
**Acknowledgments**
We thank Ke Cao, Sijing Liang, Yannan Ji, Sihe Wang for participating the field investigation.
We are grateful to the reviewers (xxxxxxxx) for their helpful comments and constructive suggestions.
This research was supported by National Key R & D Plan (Grant 2017YFC06014005), Natural
Science Foundation of China (NSFC) projects 41372106 and 41672097, and National Basic
Research Program of China (973 Project) 2012CB822003.

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






**Tables**
Table 1 Percentages of single zircons and rock samples in 1σ error (Myr), error/age ratio, and
Th/U ratio of the late Mesozoic extrusive rocks in SE China

| Sources | Con-cordant zircon Number | Rock Sample | 1σ error | | | | | | error/age | | | | | | | | Zircon Number (Th/U) | Th/U | | |
|---|---|---|---|---|---|---|---|---|---|---|---|---|---|---|---|---|---|---|---|---|
| | | | Age (Myr) | Zircon Number | % | Age (Myr) | Sample Number | % | Ratio | Zircon Number | % | Age (Myr) | Sample Number | % | | | | | Ratio | Zircon Number | % |
| This work in SHTZ | 636 | 48 | <3 | 570 | 89.6 | <2 | 46 | 95.8 | 0-3 | 581 | 91.4 | <2 | 41 | 85.4 | 636 | <0.1 | 1 | 0.2 |
| | | | 3-5 | 63 | 9.9 | 2-4 | 2 | 4.2 | 3-5 | 50 | 7.9 | 2-4 | 7 | 14.6 | | 0.1-1.0 | 20 | 3.1 |
| | | | >5 | 3 | 0.5 | >4 | | | >5 | 5 | 0.8 | >4 | | | | 1.0-10 | 615 | 96.7 |
| | | | | | | | | | | | | | | | | >10 | 0 | 0.0 |
| Composed in SHTZ | 2593 | 188 | <3 | 2066 | 79.7 | <2 | 153 | 81.4 | 0-3 | 2212 | 85.3 | <2 | 168 | 89.4 | 2503 | <0.1 | 1 | 0.0 |
| | | | 3-5 | 441 | 17.0 | 2-4 | 31 | 16.5 | 3-5 | 348 | 13.4 | 2-4 | 18 | 9.6 | | 0.1-1.0 | 945 | 37.8 |
| | | | >5 | 86 | 3.3 | >4 | 4 | 2.1 | >5 | 33 | 1.3 | >4 | 2 | 1.1 | | 1.0-10 | 1543 | 61.6 |
| | | | | | | | | | | | | | | | | >10 | 14 | 0.6 |
| Composed in SHTZ+CZ | 4639 | 291 | <3 | 3543 | 76.4 | <2 | 246 | 84.5 | 0-3 | 3798 | 81.9 | <2 | 264 | 90.7 | 4175 | <0.1 | 1 | 0.0 |
| | | | 3-5 | 898 | 19.4 | 2-4 | 39 | 13.4 | 3-5 | 769 | 16.6 | 2-4 | 25 | 8.6 | | 0.1-1.0 | 1766 | 42.3 |
| | | | >5 | 198 | 4.3 | >4 | 6 | 2.1 | >5 | 73 | 1.6 | >4 | 2 | 0.7 | | 1.0-10 | 2394 | 57.3 |
| | | | | | | | | | | | | | | | | >10 | 14 | 0.3 |

Notes: Numbers of evaluated zircon grains differ from sources in U-Pb age and Th/U ratio due to unavailability of some original dada. CZ, Coastal zone;






**Figures**

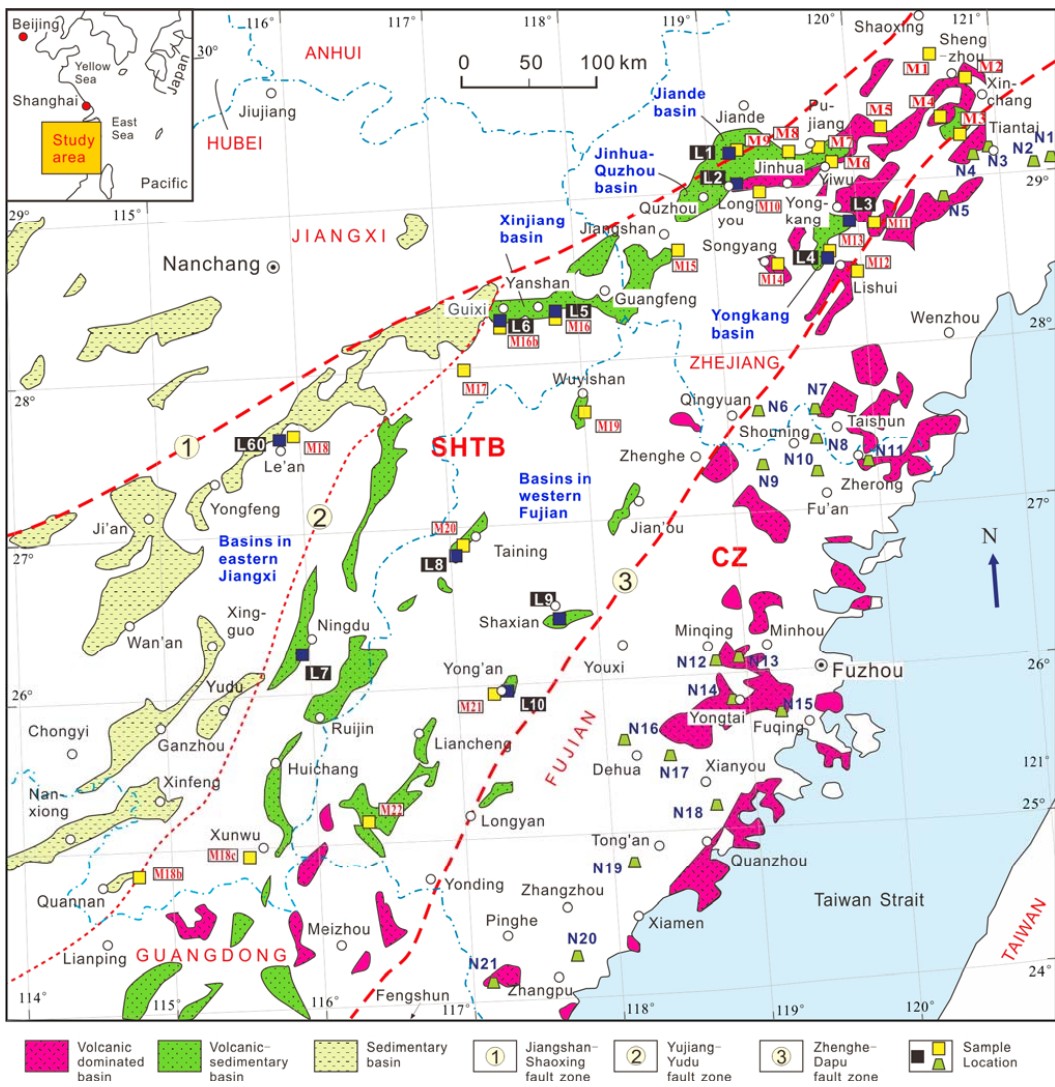


Figure 1   Sketch geological map of South China showing tectonic and basin zonations of the upper
Mesozoic and sample locations (map simplified after Shu et al., 2009). In SHTB (Shi-Hang tectonic
belt), dark blue squares with white capital letter L + numbers within black rectangles mark the
sampling locations of this study (supplementary data figures RD1, RD2, and RD3), and yellow
squares with red capital letter M + numbers within white rectangles indicate sampling locations of
previous studies (supplementary data Table RD1 and RD2). In CZ (Coastal Zone), green trapezoids
with bold capital letter N + numbers are sample locations of previous studies (supplementary data





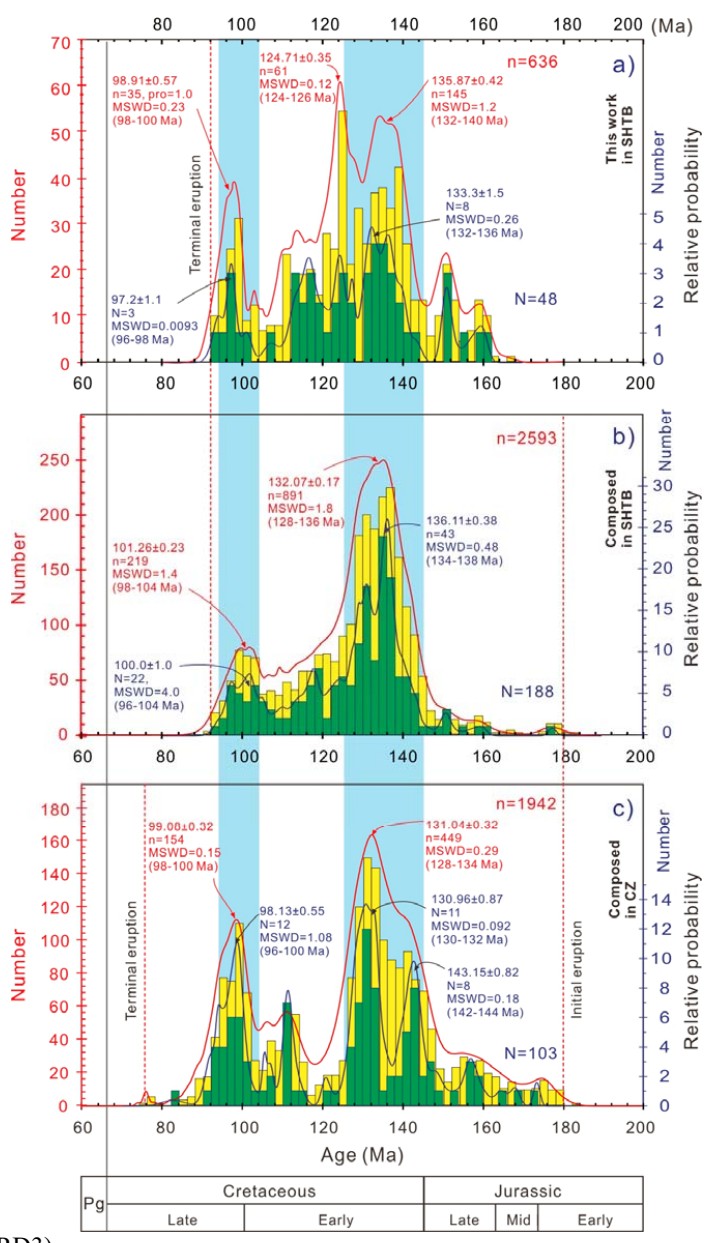

Table RD1 and RD3).

Figure 2   Relative probability and histogram diagrams of concordant zircon U-Pb isotope and
sample weighed–mean ages of extrusive rocks from SE China (details see in supplementary data
Table RD1, RD2, and RD3). a), this study in the SHTB; b), combined this and previous studies in the
SHTB; c), published data in the CZ. N = number of rock samples, n = total number of zircon grains.





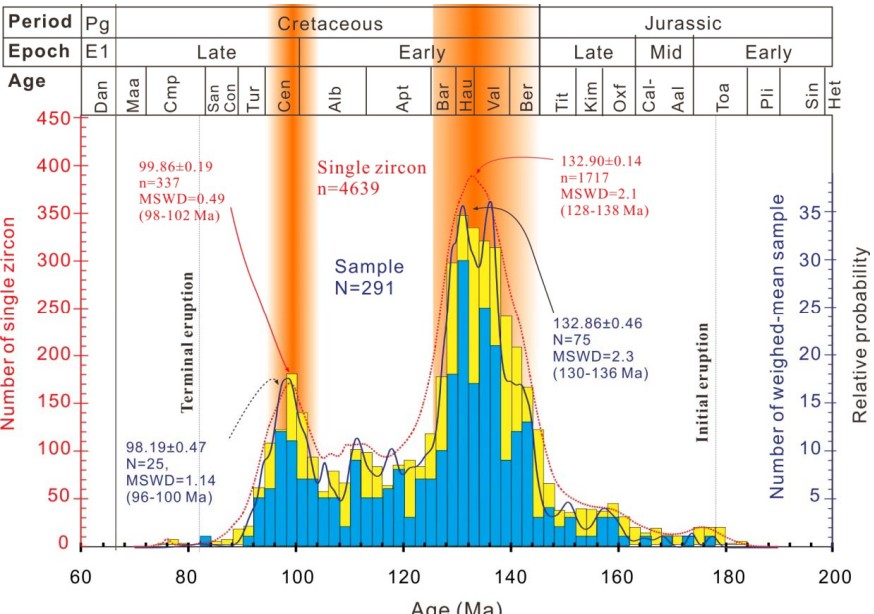

Figure 3   Diagram showing U-Pb isotope age relative probability and histogram of both single
zircon and individual sample weighed mean zircons from all extrusive rock samples in SE China. N
= number of rock samples, n = total number of zircon grains.













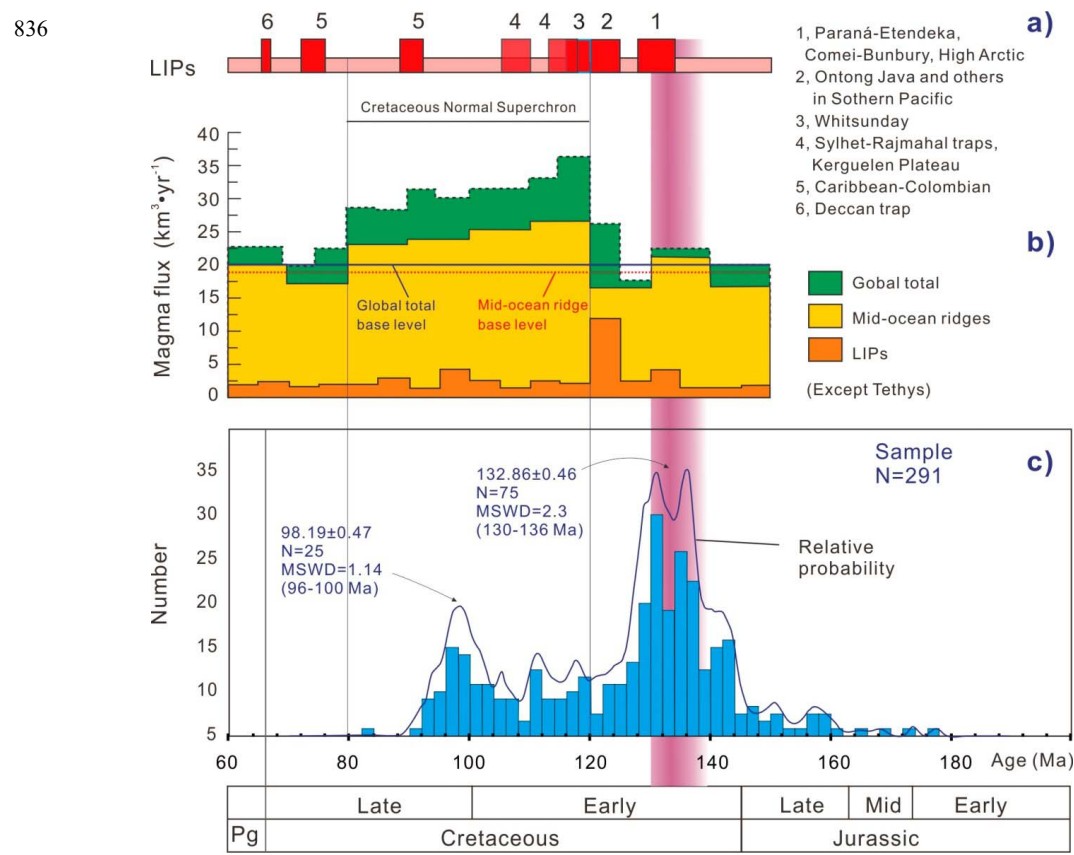

Figure 4   Diagram showing age ranges of volcanism in SE China and correlations with the global
Large Igneous Provinces (LIPs) and magmatic flux. a, age range of the Cretaceous LIPs (summary
see Coffin & Eldholm, 1994); b, magma flux of the Cretaceous LIPs, mid-ocean ridges, and (except
Tethys) global total (Coffin & Eldholm, 1994); c, age range of the volcanism with histogram and
relative probability and SE China.








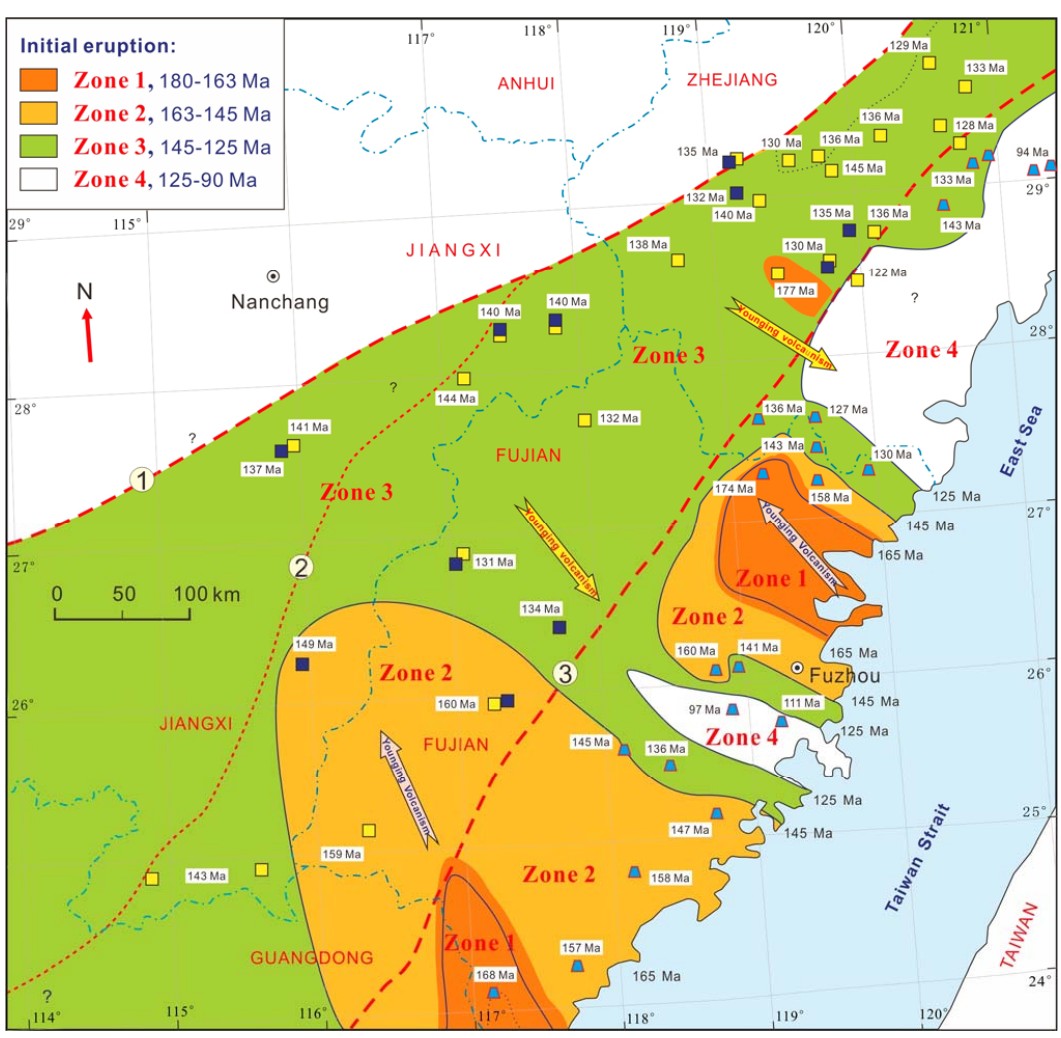

Figure 5   Sketch map showing extrusive zonations of the late Mesozoic volcanism by initial ages in SE China. Four zonations Zone **1, 2, 3,** and **4** are recognized in the order of the initial eruption age interval 177-163 Ma, 163-145 Ma, 145-125 Ma, and 125-90 Ma, separately. Age within white rectangles is the initial eruption time at a location or in a basin/region. Names of the faults, color squares and trapezoids refer to Fig. 1.




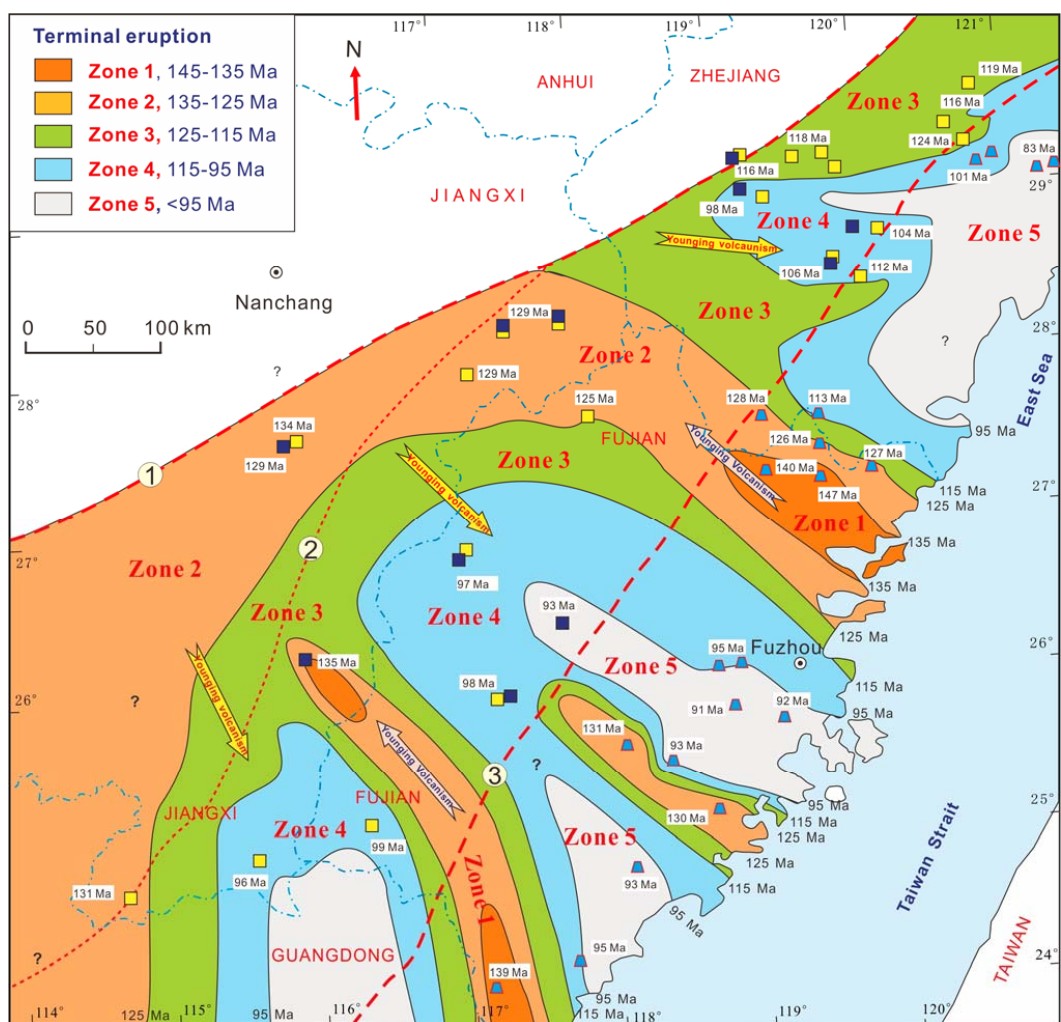


Figure 6    Sketch map showing extrusive zonations of the late Mesozoic volcanism by terminal ages
in SE China. Five zonations Zone **1, 2, 3, 4**, and **5** are recognized in the order of the terminal
eruption age interval >145 Ma, 145-125 Ma, 125-115 Ma, 115-95 Ma, and <95 Ma, separately. Age
within white rectangles is the terminal eruption time at a location or in a basin/region. Names of the
faults, color squares and trapezoids refer to Fig. 1.







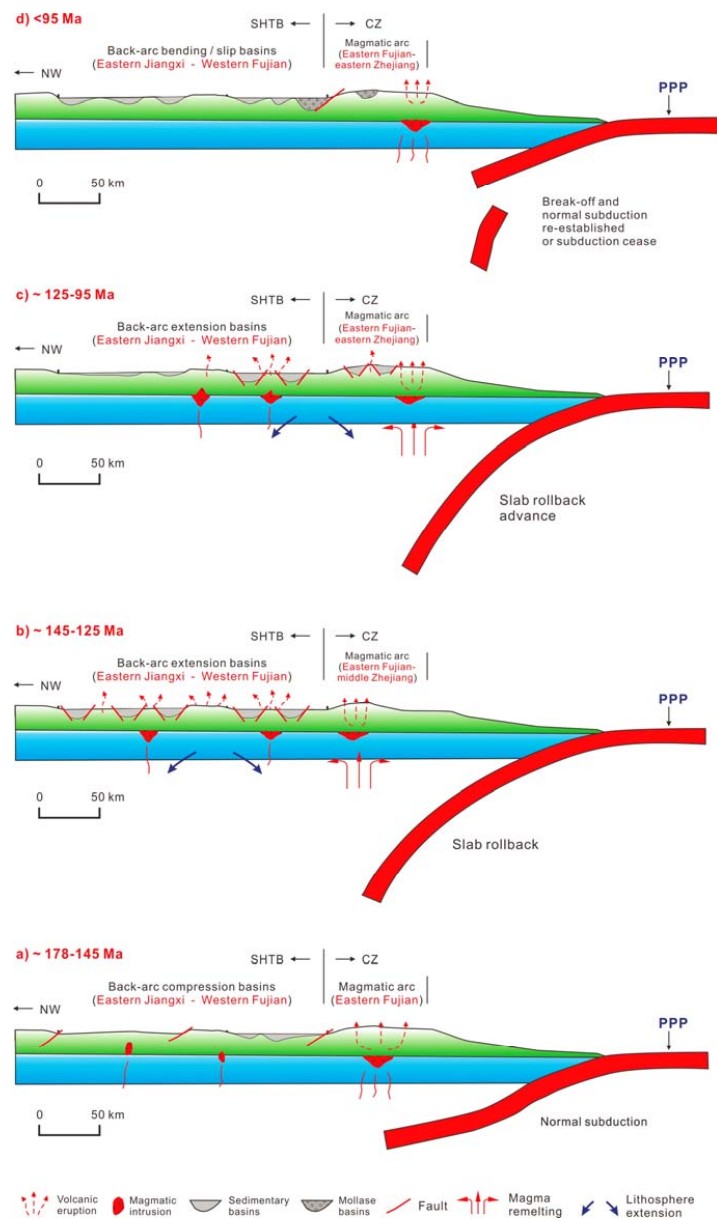

Figure 7    Cartoons showing models of the late Mesozoic volcanic advance–retreat and PPP
subduction-rollback. a) PPP normal subduction under SE China during the Middle-Late Jurassic
(178-145 Ma), during which volcanism chiefly occurred in southern and northeastern Fujian, the
magmatic arc (CZ). b) Rollback of the PPP frontier during the early Early Cretaceous (145-125 Ma),
leading to the westward volcanism and lithosphere extension in eastern Jiangxi, western Fujian, and
middle Zhejiang, back arc (SHTB). c) Rollback advance of the PPP during the late Early Cretaceous
(125-95 Ma), resulting in the eastward retreat of volcanism to Fujian and eastern Zhejiang
(SHTB-CZ). d) Re-established normal subduction or cease by break-off of the PPP after 95 Ma,
fading the magmatism in eastern Fujian and eastern Zhejiang (CZ).