# Peer review of "Tempo-spatial variation of the late Mesozoic volcanism in Southeast"

_Solid Earth, 2019_

## Referee Comment (RC1) · Anonymous Referee #1 · 11 Jun 2019

This article reports 48 new and robust U-Pb age dates for Mesozoic volcanic rocks from South China. I highly support its publication in a highly modified form.

While the data are plentiful and important, the interpretation of the data is presented in a non-scientific fashion, so the logic behind the conclusions and the importance of the data are lost. The combined dataset shows a spectrum of dates that range mostly from ca. 150-160 Ma to 90 Ma with one well defined peak at 132.9 Ma and another subordinate peak at 99.9 Ma. A bimodal distribution in ages seems robust, but this simple message is lost by (1) trying to divide South China into sub-blocks and examining age distributions independently by block, (2) too finely discussing potential

age-modes (up to five), and (3) over-interpreting the spatial and temporal distribution of the ages in a tectonic reference frame.

(1) Splitting South China into two tectonic terrains, SHTB and CZ, for a Mesozoic problem is highly questionable. There is no difference in Nd model ages between the two, suggesting their basement histories are identical. If there is a difference, then it was inherited long prior to the Mesozoic, so the designation has little meaning for the present study. Like the authors show themselves in Figs 2a and 2b, there is no difference in the general age spectra between the two. I recommend taking this out completely, otherwise you need to justify it.

(2) I cannot understand why the ages were separated into five discrete populations. Is there any statistical reason for this? If you insist on so finely interpreting the data, then you need to have statistical grounds to defend it. If one had five dates that were distinct by 1 Ma each, could this be used to argue for five separate age populations? How much data are needed before designating that a single age population has meaning? What seems robust to me is that when enough (N = ?) data are acquired, one very robust peak appears at ca. 133 Ma and a secondary one at ca. 100 Ma. How much is this conclusion biased by arbitrary (geologically speaking) sample selection? What if you obtained 48 new dates from other basins? Applying more rigorous statistical methods would significantly improve the paper as a scientific contribution.

Since southeast China had a basin and range-type setting in the Mesozoic, it seems likely the basins would preserve extrusive volcanic rocks, whereas these rocks would be eroded at the horsts (positive topography) to reveal the subterranean feeders (magma chambers) that fed the extrusive rocks. If true, can one also use the dates from the plutonic rocks to get an even better idea of the age distribution of volcanism + plutonism in South China? It would be very interesting if you could compare the two (volcanic vs. plutonic rocks) age distributions.

I highly recommend that the authors examine the data, besides just the ages, (e.g.,

U and Pb concentrations, etc.) in more detail. Are there any trends in these variable trough time/place?

(3) Figures 5, 6 and 7, together with the text in sections 5.2 and 5.3, should be omitted. It completely detracts from the important message of the paper and has no justification.

---

## Referee Comment (RC2) · Anonymous Referee #2 · 4 Jul 2019

The manuscript does a good effect in reviewing the Late Mesozoic volcanic rocks in SE China. I am glad to recommend it to be published on Solid Earth. Meanwhile I hope to point out several issues and suggestions that may help the authors improve their manuscript.

(1) There do exist some studies using zircon U-Pb ages to construct tectonic model for SE China during the Late Mesozoic time. They should be introduced in the first part of manuscript. (2) Systematic review and analysis of the chronological data across the whole SE China is necessary to build a good model for tectonic and magmatic evolution of the study region. However, it does not mean putting all zircon U-Pb ages

together. Some previous studies, focusing on individual regions, are actually trying to show the "diachronous" evolutionary trend. (3) The configurations and processes of the active continental margin appear to be constrained simply by chronological data. I suggest some geochemical data should be appended to give further constraints. For instance, why some volcanic (4) Some studies connect the Late Mesozoic tectono-magmatic processes to the Early Mesozoic ones in South China. Show the reasons to separate the two orogenies (Indosininan and Yanshanian orogenies). (5) The authors argue previous studies only involve a limited geographic region and compile data from more areas. However, show the region why Yangtze Block and western Taiwan is omitted from this study. (6) In the coastal region of Zhejiang University, there are many Early Cretaceous intermediate intrusive rocks, which have good age constraints. The authors may consider discussing these data.

A few specific comments:

Properly use hyphen and dash in the text. Check through the whole text. Use proper decimal places for zircon U-Pb ages.

Line 24: show what are CZ and CHTB first. Line 90: show the reference or evidence for the starting time. Line 286: $74.0 \pm 0.6$ Figure 7: show the vertical scale.

---

## Author Comment (AC1) · 17 Aug 2019

Reply to Referee #1

We appreciate referee #1 for the many constructive comments and helpful suggestions. Below are point-to-point replies to the comments.

-While the data are plentiful and important, the interpretation of the data is presented in a non-scientific fashion, so the logic behind the conclusions and the importance of the data are lost. The combined dataset shows a spectrum of dates that range

mostly from ca. 150-160 Ma to 90 Ma with one well defined peak at 132.9 Ma and another subordinate peak at 99.9 Ma. A bimodal distribution in ages seems robust, but this simple message is lost by (1) trying to divide South China into sub-blocks and examining age distributions independently by block, (2) too finely discussing potential age-modes (up to five), and (3) over-interpreting the spatial and temporal distribution of the ages in a tectonic reference frame.

-(1) Splitting South China into two tectonic terrains, SHTB and CZ, for a Mesozoic problem is highly questionable. There is no difference in Nd model ages between the two, suggesting their basement histories are identical. If there is a difference, then it was inherited long prior to the Mesozoic, so the designation has little meaning for the present study. Like the authors show themselves in Figs 2a and 2b, there is no difference in the general age spectra between the two. I recommend taking this out completely, otherwise you need to justify it.

**Reply:**

We agree that there is no difference in Nd model ages between the two tectonic-basin units and both could have the same basement histories in the late Mesozoic. However, the SHTB (rifting strike-slipping / back-arc) and CZ (magmatic arc) have been widely used in the literature (e.g., Gilder et al., 1996; Chen et al., 2005; Shu et al., 2009; Jiang et al., 2011; Yang et al., 2012) . Therefore, we kept the SHTB and CZ description in both the "Geological setting" and the "Results" sections, and in Figure 1. We have deleted them in the "Discussion" section and figure 5 (age spatial distribution map) in the revised version.

-(2) I cannot understand why the ages were separated into five discrete populations. Is there any statistical reason for this? If you insist on so finely interpreting the data, then you need to have statistical grounds to defend it. If one had five dates that were distinct by 1 Ma each, could this be used to argue for five separate age populations? How much data are needed before designating that a single age population has meaning? What
seems robust to me is that when enough (N = ?) data are acquired, one very robust peak appears at ca. 133 Ma and a secondary one at ca. 100 Ma. How much is this conclusion biased by arbitrary (geologically speaking) sample selection? What if you obtained 48 new dates from other basins? Applying more rigorous statistical methods would significantly improve the paper as a scientific contribution.

**Reply:**

We thank the reviewer for the constructive suggestion and advice. We did not discuss too finely about the age-models in the revision.

-Since southeast China had a basin and range-type setting in the Mesozoic, it seems likely the basins would preserve extrusive volcanic rocks, whereas these rocks would be eroded at the horsts (positive topography) to reveal the subterranean feeders (magma chambers) that fed the extrusive rocks. If true, can one also use the dates from the plutonic rocks to get an even better idea of the age distribution of volcanism + plutonism in South China? It would be very interesting if you could compare the two (volcanic vs. plutonic rocks) age distributions.

**Reply:**

In the late Mesozoic, the Shi-Hang tectonic belt (SHTB) in SE China is the zone that the rifting or back arc basin occupied, and the Coastal zone (CZ) is the region that range-type spread. It is sure that both of the basins (SHTB) and ranges (CZ) have preserved great numbers of extrusive volcanic rocks. It is extremely probable that the range-type (horst) volcanic rocks could have been eroded and the exposed plutonic rocks could have fed the basins. It is a great suggestion to combine the volcanic rocks with plutonic rocks to analyze the age distribution for the late Mesozoic volcanism in SE China. But some new issues would produce if we do as suggested.

Firstly, the revealed plutonic rocks in horst (CZ) could be mostly earlier than the extrusive rocks in basin (SHTB), because these plutons must have intruded into the host
rocks earlier than the covered extrusive rocks. Those coevally erupted pyroclastic matters and ashes related to plutonic magma would have already been fallen in basin and / or range in advance, and the revealed plutonic rocks must have fed the basins with weathered and transported terrigenous clastics/fragments when the covered rock are eroded.

Secondly, the probability distribution of the ages will become more complicated and even lead to the misunderstanding of the abundance as plutonic rocks could be different in age. Supposed that the revealed plutonic rocks are coeval with the extrusive rocks preserved in basins, the age distribution would become mixed and confused if we plot the ages of two kinds of rocks on the map.

Thirdly, this work focused on the volcanic eruption, and it would take much time and lots of pages to embody the extra rocks if we consider intrusive rocks in SE China. Another problem may arise if we do, that is, plutonic rocks are not representative as the extrusive rocks in age because some chamber rocks may have not been uplifted and exposed to air.

-I highly recommend that the authors examine the data, besides just the ages, (e.g., U and Pb concentrations, etc.) in more detail. Are there any trends in these variable trough time/place?

Reply:

Thank you for the high recommendation! We have carefully examined the original data of zircon U-Pb dating.

As you can see from the supplementary data Table RD2 and RD3 (zircon U-Pb isotope dating data in whole SE China), we not only carefully examined our data but also checked the concentrations and ratio of single zircon U and Pb isotopes from the cited references one by one. Particularly, we have marked and got rid of those ages with > 5% age error and abnormal U and Pb concentration and ratio that were used in those
original references. In summary, we have been following the lab and data regulations (e.g., Klötzli et al., 2009; Solari et al., 2010; Li et al., 2015) of the precision and accuracy uncertainties. A figure is enclosed as attachment, showing the examination results by comparison of relative probability and histogram between single zircon 206Pb/238U ages and Th/U ratios. The examination results demonstrate the same pattern of the weighed-mean ages by rock sample.

Therefore, there are no artificial trends in age and place for both single zircons and rock samples.

-(3) Figures 5, 6 and 7, together with the text in sections 5.2 and 5.3, should be omitted. It completely detracts from the important message of the paper and has no justification.

Reply:

While it is an over-interpretation on model of the Paleo-Pacific Plate subduction by the spatial and temporal distribution of the ages from South China, the tempo-spatial variations represent an important observation of this study. So, we deleted figure 7 and section 5.3 in the revised version, and the title has been changed to "Tempo-spatial variation of the late Mesozoic volcanism in Southeast China". We combined the previous figure 5 with figure 6 as the new figure 5. We have also made modification of both figures and text accordingly.

Please also note the supplement to this comment: https://www.solid-earth-discuss.net/se-2019-76/se-2019-76-AC1-supplement.pdf

SED
Comparative diagram of relative probability and histogram between single zircon  $^{206}Pb/^{238}U$  ages and Th/U ratios of extrusive rock samples in SE China. A, zircon  $^{206}Pb/^{238}U$  ages with 1 Myr bin; B,  $^{206}Pb/^{238}U$  ages with 2 Myr bin; C, zircon Th/U ratios with 1 Myr bin; D, zircon Th/U ratios with 2 Myr bin. N = number of rock samples, n = total number of zircon grains.

Fig. 1.

---

## Author Comment (AC2) · 17 Aug 2019

We thank Referee #2 for the critical issues and constructive suggestions. Below are replies to questions one by one.

————————————————-

The manuscript does a good effect in reviewing the Late Mesozoic volcanic rocks in SE China. I am glad to recommend it to be published on Solid Earth. Meanwhile I hope to point out several issues and suggestions that may help the authors improve their manuscript.

[Figure]

–(1) There do exist some studies using zircon U-Pb ages to construct tectonic model for SE China during the Late Mesozoic time. They should be introduced in the first part of manuscript.

Reply:

Following Referee #1's suggestion, we have removed this topic in the revision.

–(2) Systematic review and analysis of the chronological data across the whole SE China is necessary to build a good model for tectonic and magmatic evolution of the study region. However, it does not mean putting all zircon U-Pb ages together. Some previous studies, focusing on individual regions, are actually trying to show the "diachronous" evolutionary trend.

Reply:

Yes, there some previous studies at individual regions that suggest "diachronous" evolution of volcanism. However, for these studies, it is hard to distinguish whether these features represent local or regional signature. To avoid this ambiguity, we analyze the chronological data from the entire SE China and investigate the temporal-spatial variations of volcanism with the age data from SE China.

–(3) The configurations and processes of the active continental margin appear to be constrained simply by chronological data. I suggest some geochemical data should be appended to give further constraints. For instance, why some volcanic (authors note: hereafter some words are missed).

Reply:

Geochemical data would be very helpful for constraining the tectonic model. Following Referee #1's suggestion, we have removed the discussion of tectonic model in the revision and focused on the temporal-spatial variations of volcanism in SE China.

–(4) Some studies connect the Late Mesozoic tectonomagmatic processes to the Early

[Figure]

Mesozoic ones in South China. Show the reasons to separate the two orogenies (In-dosininan and Yanshanian orogenies).

Reply:

Traditionally, the Indosinian and Yanshanian orogenies can be distinguished by time in East and Southeast Asia i.e., Indosinian orogeny took place during the Late Triassic-Early Jurassic, and the Yanshanian orogeny happened during the Late Jurassic-Cretaceous. All ages (177-82 Ma) measured in this work and cited from references from study area (see supplementary data Table RD1) show they belonged to the Yan-shanian orogeny.

The other criterion is the tectonic disconformity contact of the strata. In SE China, there are few volcanic and / or volcanic-sedimentary records formed during the Late Triassic-Early Jurassic, but abundant extrusive rocks are well preserved in the Upper Jurassic – Cretaceous strata. Actually, unconformable contacts were already described and discussed between the strata which were formed during the Indosinian and Yanshanian orogenies, respectively (summary see in Shu LS et al., 2009, JAE; Li XH et al., 2019, GR), for which it seems dispensable to repeat reasons separating the two orogenies in this paper. All the samples are from the Upper Jurassic – Cretaceous strata in SHTB, in both this work and those citations. For details of samples and strata please refer to the supplementary data (Table RD1).

–(5) The authors argue previous studies only involve a limited geographic region and compile data from more areas. However, show the region why Yangtze Block and western Taiwan is omitted from this study.

Reply:

Our study focused on the late Mesozoic volcanism in SE China that occurred in the Shi-Hang tectonic belt (SHTB) and Coastal (magmatic arc) zone (CZ).

There are few volcanic rocks recorded in the late Mesozoic strata within Yangtze Block,

though sporadic coeval volcanic rocks were reported in the Lower Yangtze area (Lu-zong and Ningwu. e.g., Deng et al., 2012, Int. Geol. Rev.; Chen et al., 2014, Lithos; Liu et al., 2014, J. Geochem. Exp.). They have few contributed to the volcanism in SE China, and may not be attributed to the PPP subduction. So did those in Hunan and Hubei in middle South China (Middle Yangtze Block).

There are no volcanic rock materials available in western Taiwan Island and Taiwan Strait, at least, no late Mesozoic volcanic rocks were reported until present. In eastern Taiwan, the Cretaceous-Cenozoic fore-arc basin has been developed without mag-matic occurrence (report), leading to no citation of volcanism.

–(6) In the coastal region of Zhejiang University (Authors note: "University" should be province), there are many Early Cretaceous intermediate intrusive rocks, which have good age constraints. The authors may consider discussing these data. This paper focuses on the volcanic rocks. It could become more complicated if combine with intrusive rocks in local place even in the whole SE China. Please see the similar explanation in third part (3).

–A few specific comments: Properly use hyphen and dash in the text. Check through the whole text. Use proper decimal places for zircon U-Pb ages.

Reply:

We have paid much attention for the usage of hyphen and revised accordingly in the revised version.

–Line 24: show what are CZ and SHTB first. Line 90: show the reference or evidence for the starting time. Line 286: 74.0 _ 0.6 Figure 7: show the vertical scale.

Reply:

Line 24: We have used the original words at the first appearance of the CZ and SHTB. Line 90: The sentence was rewritten and references are added (e.g., Wang et al., 2013). Line 286: It was corrected as 74.0 $\pm$ 0.6 Ma. Figure 7: This figure was deleted

in the new version following the suggestion by the referee #1.

Please also note the supplement to this comment:
https://www.solid-earth-discuss.net/se-2019-76/se-2019-76-AC2-supplement.pdf

---

## Author Response (AR1)

- 1 Tempo-spatial variation of the late Mesozoic volcanism in Southeast
- 2 China-testing the western Paleo-Pacific Plate subduction models
- Authors' Note: Following Referee #1, the implication of tectonic is over-interpreted. Then we deleted the relevant
  phrase within title.
- 6 7 **Xianghui Li1, 2\* Yongxiang Li1 Jingyu Wang1 Chaokai Zhang1 Yin Wang3 Ling Liu3**
- 1State Key Laboratory for Mineral Deposits Research, School of Earth Sciences and Engineering,
   Nanjing University, Nanjing 210023 China. Email: leeschhui@126.com
- 10 2State Key Laboratory of Oil and Gas Reservoir Geology and Exploitation, Chengdu University of
- 11 Technology, Chengdu 610059, China
- 12 3East China Mineral Exploration and Development Bureau, Nanjing 210007, China
- 13

**14 ABSTRACT**

The magmatism (including volcanism) in East Asia (/ China) could provide key clues and age 15 16 constrains for the subduction and dynamical process westward subduction of the Paleo-Pacific plate (PPP) played a governing role in tectonic evolution of East Asia. Although lots of absolute isotope 17 18 ages of extrusive rocks have been published in the 1980s-2000s, large uncertainties and big errors 19 prevent the magmatism in SE China from being well understoodvarious PPP subduction models have been proposed, the subduction age and dynamical process of the PPP remain controversial. In this 20 21 study, we investigate the zircon geochronology of extrusive rocks and tempo-spatial variations of the 22 late Mesozoic volcanism in Southeast (SE) China. We reported zircon U-Pb ages of new 48 extrusive rock samples in the Shi-Hang tectonic zone. Together with the published data in recent decades, ages 23 of  $\sim 300291$  rock samples from  $\sim 40$  lithostratigraphic units were compiled, potentially documenting a 24 25 relatively complete history and spatial distribution of the late Mesozoic volcanism in Southeast-SE 26 China. The results show that the extrusive rocks spanned ~95 Myr (177-82 Ma), but dominantly ~70 27 Myr (160-90 Ma), within which with two main the volcanism in the early Early Cretaceous (age populations of 145-125 Ma) iwas the most intensive and widespread eruption and 105-95 Ma. We 28 29 propose that these ages represent the intervals of the Yanshanian volcanism in Southeast SE China and the western subduction of the PPP, within which two intensive volcanic eruptional pulses 30 31 happened... Spatially, the age geographic pattern of extrusive rocks showis that both the oldest and youngest age clusters occur in coastal magmatic arc (eastern Zhejiang the CZ and Fujian), and the 32 younger-most intensive and widespread age group (145-125 Ma) occurs in the SHTBback arc / 33 34 rifting basin (eastern Jiangxi, middle Zhejiang, and northern Guangdong), indicating-implying that the late Mesozoic volcanism migrated northwestly from the coast to the inland prior to ~145 Ma and 35 subsequently retreated southeastly back to the coast. - This volcanic migration pattern is interpreted 36 37 to result may imply that the Paleo-Pacific plate subducted northwestward and the roll-back subduction did not begin until the Aptian (~125 Ma) of the mid-Cretaceousfrom a northwestward 38

- 39 subduction followed by a southeastward rollback or retreat of the PPP.
- 40 Authors' Note: Some introductions on tectonic implications were deleted following the suggestion by Referee #1.
- 41 Keywords: geochronology; tempo-spatial variation; volcanism; late Mesozoic; Southeast China;
- 42 Paleo-Pacific Plate
- 43

**44 **1. Introduction**

[revised manuscript text omitted]
 corner of Fujian, due similar to only those-one location of the terminal ageof the initial age distribution. Zone 2 (135145-125 Ma) mainly occurs in eastern Jiangxi 418 and, western SHTB while partly surrounds the Zone 1 banded boundary of northern Fujian; Zone 3 (125-105-100 Ma) largely distributes in the boundary region of eastern Jiangxi and western Fujian 419 and in middle and southwestern Zhejiang in the SHTB. Zone 4 (105100-95-83 Ma) widely appears in 420 421 regions of the southern-middle Fujian, middle eastern Zhejiang-in the eastern SHTB and CZ, and northern Guangdong. Zone 5 (<95 Ma) sporadically displays in the eastern Fujian, eastern Zhejiang, 422 and northern Guangdong in the CZ. Same imilar zonations can be classified in the map sketched by 423 424 the single zircon U-Pb ages (supplementary data Fig. RD14), verifying the zones of the sample weighed-mean ages in SE China. 425

Zonations of both-initial, peak, -and terminal volcanism indicate a distinct pattern of volcanic 426 extrusion in SE China (Figs. 5-and 6): the oldest ages in the Ceastern SE ChinaZ, the younger 427 intensive age clusters in the SHTB western SE China, and the youngest ones in eastern SE China 428 429 againthe CZ. Detailed distributional patterns can be observed: 1) the earliest appearance and earliest 430 disappearance of extrusive rocks dominantly occur in southeastern and northeastern Fujian, where the magmatic arc was located (e.g., Lapierre et al., 1997) in the CZ; 2) the most widespread 431 distribution of <del>145-125 Ma</del> extrusive rocks <del>are</del> is the most intensive volcanism age as 145-125 Ma in 432 433 eastern Jiangxi, western middle Zhejiang, and western Fujiannorthern Guangdong, in which a back-arc / rifting basin was developed (e.g., Gilder et al., 1991; Jiang et al., 2009; 2011) in the SHTB; 434 3) the latest appearance and latest disappearance mainly occur in eastern Zhejiang, and eastern 435 Fujian in the CZ, and northern Guangdong. 436

With the observation of volcanism, two distributional patterns manifest: 1) the migration of the volcanism was from the northwestward to the southeastward, implying that the PPP could have been subducted northwestly during the late Mesozoic time; 2) the first appearance (initial volcanism) area and is-the first disappearance (terminal volcanism) region are the same region, suggesting that a roll-back subduction of the PPP happened after ~125 Ma.

442

It is surprising that the zone 1 and / or 2 of both initial and terminal-volcanism look like

thermal-dome patterns (Fig. 5-and 6) by exhumation and exposure that may be related to the regional
magmatic intrusion, likely misleading the migration of volcanism. However, the distribution pattern
is not dome-controlled because: 1) The data are derived from extrusive rocks, instead of intrusive
rocks; 2) it is impossible that a crater is over 200-300 km wide in diameter; 3) lots of agglomerates
representing craters were observed in a variety of strata at locations / basins out of Zone 1. For
instance, these agglomerates are widespread in basins of western Zhejiang (L1~L4; M9~M14),
eastern Jiangxi (L5~L7; M16~M18b), and western Fujian (L8~L10, M19~M22).

450 Authors' Note: The below section 5.3 on tectonic implication was completely deleted following the suggestion by

451 Referee #1.

[revised manuscript text omitted]

|           | Con-
cordant
zircon
Number | Rock
Sample | 1 o error        |                  |      |              |                      |      | error/age |                  |      |              |                  |      | Zircon           | Th/U    |                  |      |
|-----------|-------------------------------------|----------------|------------------|------------------|------|--------------|----------------------|------|-----------|------------------|------|--------------|------------------|------|------------------|---------|------------------|------|
| Sources   |                                     |                | Age
(Myr
) | Zircon
Number | %    | Age
(Myr) | Sample
Numbe
r | %    | Rati
o | Zircon
Number | %    | Age
(Myr) | Sample
Number | %    | Number
(Th/U) | Ratio   | Zircon
Number | %    |
| This work | 636                                 | 48             | <3               | 570              | 89.6 | <2           | 46                   | 95.8 | 0-3       | 581              | 91.4 | <2           | 41               | 85.4 | 636              | <0.1    | 1                | 0.2  |
| in SHTZ   |                                     |                | 3-5              | 63               | 9.9  | 2-4          | 2                    | 4.2  | 3-5       | 50               | 7.9  | 2-4          | 7                | 14.6 |                  | 0.1-1.0 | 20               | 3.1  |
|           |                                     |                | >5               | 3                | 0.5  | >4           |                      |      | >5        | 5                | 0.8  | >4           |                  |      |                  | 1.0-10  | 615              | 96.7 |
|           |                                     |                |                  |                  |      |              |                      |      |           |                  |      |              |                  |      |                  | >10     | 0                | 0.0  |
| Composed  | 2593                                | 188            | <3               | 2066             | 79.7 | <2           | 153                  | 81.4 | 0-3       | 2212             | 85.3 | <2           | 168              | 89.4 | 2503             | < 0.1   | 1                | 0.0  |
| in SHTZ   |                                     |                | 3-5              | 441              | 17.0 | 2-4          | 31                   | 16.5 | 3-5       | 348              | 13.4 | 2-4          | 18               | 9.6  |                  | 0.1-1.0 | 945              | 37.8 |
|           |                                     |                | >5               | 86               | 3.3  | >4           | 4                    | 2.1  | >5        | 33               | 1.3  | >4           | 2                | 1.1  |                  | 1.0-10  | 1543             | 61.6 |
|           |                                     |                |                  |                  |      |              |                      |      |           |                  |      |              |                  |      |                  | >10     | 14               | 0.6  |
| Composed  | 4639                                | 291            | <3               | 3543             | 76.4 | <2           | 246                  | 84.5 | 0-3       | 3798             | 81.9 | <2           | 264              | 90.7 | 4175             | < 0.1   | 1                | 0.0  |
| in        |                                     |                | 3-5              | 898              | 19.4 | 2-4          | 39                   | 13.4 | 3-5       | 769              | 16.6 | 2-4          | 25               | 8.6  |                  | 0.1-1.0 | 1766             | 42.3 |
| SHTZ+C    |                                     |                | >5               | 198              | 4.3  | >4           | 6                    | 2.1  | >5        | 73               | 1.6  | >4           | 2                | 0.7  |                  | 1.0-10  | 2394             | 57.3 |
| Z         |                                     |                |                  |                  |      |              |                      |      |           |                  |      |              |                  |      |                  | >10     | 14               | 0.3  |

Table 1 Percentages of single zircons and rock samples  $-in 1\sigma$  error (Myr), error/age ratio, and Th/U ratio of the late Mesozoic extrusive rocks in SE China

899 Notes: Numbers of evaluated zircon grains differ from sources in U-Pb age and Th/U ratio due to unavailability of some original dada. CZ, Coastal zone;
 900

**902 Figures**

903